# Detecting Abnormal Axillary Lymph Nodes on Mammograms Using a Deep Convolutional Neural Network

**DOI:** 10.3390/diagnostics12061347

**Published:** 2022-05-29

**Authors:** Frederik Abel, Anna Landsmann, Patryk Hejduk, Carlotta Ruppert, Karol Borkowski, Alexander Ciritsis, Cristina Rossi, Andreas Boss

**Affiliations:** Department of Diagnostic and Interventional Radiology, University Hospital Zurich, University of Zurich, 8091 Zurich, Switzerland; frederik.abel@usz.ch (F.A.); anna.landsmann@usz.ch (A.L.); patryk.hejduk@usz.ch (P.H.); carlotta.ruppert@googlemail.com (C.R.); karol.borkowski@usz.ch (K.B.); alexander.ciritsis@usz.ch (A.C.); cristina.rossi@usz.ch (C.R.)

**Keywords:** mammography, axillary lymph nodes, suspicious lymph nodes, breast cancer, mammography screening, dCNN, deep learning, artificial intelligence

## Abstract

The purpose of this study was to determine the feasibility of a deep convolutional neural network (dCNN) to accurately detect abnormal axillary lymph nodes on mammograms. In this retrospective study, 107 mammographic images in mediolateral oblique projection from 74 patients were labeled to three classes: (1) “breast tissue”, (2) “benign lymph nodes”, and (3) “suspicious lymph nodes”. Following data preprocessing, a dCNN model was trained and validated with 5385 images. Subsequently, the trained dCNN was tested on a “real-world” dataset and the performance compared to human readers. For visualization, colored probability maps of the classification were calculated using a sliding window approach. The accuracy was 98% for the training and 99% for the validation set. Confusion matrices of the “real-world” dataset for the three classes with radiological reports as ground truth yielded an accuracy of 98.51% for breast tissue, 98.63% for benign lymph nodes, and 95.96% for suspicious lymph nodes. Intraclass correlation of the dCNN and the readers was excellent (0.98), and Kappa values were nearly perfect (0.93–0.97). The colormaps successfully detected abnormal lymph nodes with excellent image quality. In this proof-of-principle study in a small patient cohort from a single institution, we found that deep convolutional networks can be trained with high accuracy and reliability to detect abnormal axillary lymph nodes on mammograms.

## 1. Introduction

Female breast cancer (BC) is the most frequently diagnosed cancer globally, with 2.3 million new cases (11.7% of all cancers combined) each year, followed by lung cancer (11.4%). In women, breast cancer remains the leading cause of cancer death [1].

Since implementation in various countries, mammography screening programs are estimated to contribute to a 22% reduction of breast cancer mortality worldwide. Moreover, the effect of attending screening decreases the risk of death by approximately 30% [2]. Although screening programs in routine healthcare settings confer substantial reduction in mortality from breast cancer, the effectiveness of mammography in individual populations is affected by several factors. The performance of mammography screening significantly depends on technology and interobserver agreement [3], which often leads to high recall and false positive rates resulting in unnecessary biopsies, increased healthcare costs, and psychological distress in patients [4,5]. Considering the high workload of radiologists reading digital mammograms and at least 25% of detectable cancers being missed [6,7], it is mandatory to minimize interpretation errors.

Therefore, techniques for observer-independent detection of suspicious lesions are highly desired. Substantial improvements of diagnostic accuracy in a standardized way are provided by applying machine learning approaches, especially deep convolutional neural networks (dCNNs). In recent years, dCNNs have attracted great attention due to their outstanding performance in image pattern recognition [8]. Since mammograms are single-slice-projection images, they represent a perfect target to be evaluated by a dCNN mimicking specialized and experienced human decision-making. In mammograms, dCNNs have already been proven to detect cancer with high accuracy similar to radiologists [9]. Additionally, dCNNs have successfully been applied to accurately classify breast density and microcalcifications according to the ACR BI-RADS system [10,11]. Despite remarkable and ongoing success of artificial intelligence (AI) in medical applications, a main obstacle for successful implementation in clinical practice is that they are seen as a “black box”. Understanding how the algorithm processes input data and linking it to a final prediction is challenging [12]. Hence, explainable AI systems with transparent decision-making are essential to achieve widespread acceptance for the medical domain, ultimately leading to improved screening, diagnostics, and follow-up.

Currently, most deep learning models trained on mammography studies focus on detection of three typical abnormalities: microcalcification cluster, mass, and architectural distortions. However, another finding occasionally detected on mediolateral oblique mammographic images is abnormal axillary lymph nodes. If axillary lymphadenopathy is lacking benign features and presents homogeneously dense with loss of fatty hilum and enlarged with irregular or round shape, it is strongly associated with malignancy [13,14]. In case of primary breast cancer, axillary lymph node status remains an essential factor for prognosis and facilitating pretreatment planning [15,16]. Consequently, it is crucial to recognize these pathological lymph nodes for radiologists and reduce observational oversights on routine mammograms, especially if abnormal lymph nodes represent the only pathologic finding.

So far, no study has investigated a deep neural network for classification of abnormal axillary lymph nodes found on mammograms. Here, we tested in a retrospective cohort study whether a dCNN allows for accurate, objective, and standardized classification of breast tissue, benign, and suspicious axillary lymph nodes on mammographic images according to the corresponding radiologic reports, and thereby could serve as a quality control tool.

## 2. Materials and Methods

### 2.1. Study Population

A retrospective analysis of patient data in the local picture archiving and communication system (PACS) and report (RIS) database of our institution was performed and approved by the local ethics committee. Patients signed informed consent for scientific evaluation of the imaging and clinical data. In consideration of the American College of Radiology (ACR)-released Breast Imaging Reporting and Data System (BI-RADS), a search was performed using the following search terms: “BI-RADS 6” (known biopsy-proven malignancy), “BI-RADS 5” (highly suggestive of malignancy), and “suspicious lymph nodes”. The search yielded 107 mammograms in mediolateral oblique projections (MLO) of 74 patients from the years 2010–2020. According to the corresponding radiological reports, the cohort included 33 patients (43 mammograms) with suspicious axillary lymph nodes as visualized in MLO projection. Among the cohort, 10 patients belonged to ACR BI-RADS 6 (12 mammograms), 22 patients to BI-RADS 5 (30 mammograms), and 1 patient had a follicular lymphoma (1 mammogram), representing a cohort with high risk of suspicious/malignant axillary lymph nodes. The rest of the patients (64 mammograms) without projection of abnormal lymph nodes were included as controls. As standardized in our institution, all mammograms were double-read by two experienced radiologists in breast imaging.

### 2.2. Data Preparation

The dimensions of the mammograms were adjusted to 3506 × 2800 pixels. For labeling, a custom-made MATLAB tool (Version 9.11, The MathWorks, Natick, MA, USA) was used. Each mammogram was labeled manually by one of us (FA), according to the initial radiologic reports, into three classes, localized by rectangular region of interests (ROIs) and subsequently saved as new cropped image (351 × 280 pixels). The three classes were defined as follows: (1) “breast tissue” (normal fatty/fibroglandular tissue, no lymph nodes), (2) “benign lymph nodes” (normal size, oval shape, fatty hilum), and (3) “suspicious lymph nodes” (enlarged, irregular/round shape, non-fatty). Representative examples of the three classes are depicted in Figure 1. In order to expand the size of the dataset, image augmentation was performed by the ImageDataGenerator of Keras (Version 2.4.3; Massachusetts Institute of Technology, Cambridge, MA, USA). The program provides real-time data augmentation by generating transformations for each image and training epoch such as random rotations, flips, and shifts. This results in a similar number of transformed image copies for each class (Table 1). The images have been randomly shuffled in each class. The dataset was split: 70/20% for training of the dCNN and validation of the resulting model, respectively. Ultimately, the performance of the dCNN was evaluated on a “real world” test dataset comprising 10% of the data not previously used either for training or validation and spared from data augmentation.

### 2.3. Training of dCNN Model

Computations were performed on an desktop computer (Intel i7-9700 CPU, Intel Corporation, Santa Clara, CA, USA; 16 GB RAM; NVIDIA RTX 2080 8 GB graphics, Nvidia Corporation, Santa Clara, CA, USA). The desktop PC was running under Ubuntu Linux 20.04 with Tensorflow 2.5.0. and Keras 2.4.3. All programing was performed in the computer language Python (Version 3.8.5; Python Software Foundation, Wilmington, DE, USA).

A single dCNN model was generated classifying the above described three classes (1, “breast tissue”; 2, “benign lymph nodes”; 3, “suspicious lymph nodes”). The architecture for lymph node detection was adapted and optimized from previous projects of our group [11,17,18,19]. The dCNN was implemented in Keras/TensorFlow and designed with 12 convolutional layers followed by 3 dense layers. The batch size was set to 16, and the number of epochs for training to 120. Training and validation accuracies, model architecture, and parameters were saved.

### 2.4. Human Readout on “Real World” Data

The test dataset was divided in a subset containing 60 mammograms and presented in random order to two experienced radiologists in breast imaging (2 years and 16 years of experience). Both readers were blinded to patient information and classified each image according to the three classes individually. The initial classification of the dataset according to the radiological reports served as ground truth for the evaluation of the classification accuracy of the dCNN and the two readers. Inter-reader agreement was assessed between the dCNN and each reader as well as agreement between both readers.

### 2.5. Computation of Colored Probability Maps

Representative mammograms were analyzed by a sliding window method with a pixelwise change of the x- and y-position in two nested loops over the complete width and height of the underlying mammogram (minus 351 and 280 in the two directions, respectively), applying a custom Python script. At each (x, y) position, a 351 × 280 array was cropped and classified according to the trained dCNN model. Subsequently, the probabilities of each class were calculated by the dCNN and at the respective coordinates. The probabilities were stored for each class with numerical values ranging between 0 and 1. Once the loop had finished, the resulting probability array was visualized by assigning a heat colormap to class 3 (suspicious lymph nodes). Ultimately, overlays of the original mammogram and the heat colormap were generated by adjusting the overlay map to the size of the original mammogram of 3506 × 2800. The approximate computation time of a colored probability map for a standard mammography ranged between 3 and 5 h, depending on the mammogram.

### 2.6. Statistical Analysis

Statistical computations were performed using the computer language Python. Categorical variables were expressed as frequencies or percentages. To assess the performance of the dCNN, a confusion matrix was calculated. Additionally, the macro-F1 score and Matthew correlation coefficient (MCC) were stated. For inter-reader agreement of the human readout between the dCNN and both readers, the intraclass correlation coefficient (ICC) [20,21] was calculated and values greater than 0.90 interpreted as excellent agreement. Additionally, Cohen’s Kappa [22] coefficients were determined to assess inter-reader reliabilities for the dCNN and the readers, with values between 0.81–1.0 indicating almost perfect agreement. An a priori alpha error *p* < 0.05 was considered statistically significant.

## 3. Results

### 3.1. Accuracy of the dCNN Model

Following image preprocessing, 1477 images from 107 mammograms of the patient cohort were successfully classified into three classes. After applying the data augmentation algorithm, a single dCNN model was trained and validated with 5385 images from the cohort. Progression of the models’ accuracy is illustrated in Figure 2. Initially, the accuracy of the validation data was lower compared to the training data. Reaching a higher number of epochs, the two datasets subsequently converged. The accuracy of the model obtained its maximum at epoch 119 both for the training set (98%) and for the validation set (99%).

The test set representing “real world” data of 448 images (class 1, 202 images (45%); 2, 150 images (33.5%); 3, 96 images (21.5%)) was classified according to the radiological report serving as ground truth and subsequently was applied to the trained dCNN for evaluation of the final performance. An example of the classification probability for three representative images of the test set calculated by the dCNN is shown in Figure 3. The performance of the model in classifying the three classes is demonstrated by a confusion matrix (Figure 4), which compares the predicted values of the model with the ground truth values. The resulting accuracy is 98.51% for breast tissue, 98.63% for benign lymph nodes and 95.96% for suspicious lymph nodes, representing an overall accuracy of 98% on the “real world” test set for the three classes. Accordingly, the computed F1 score and MCC of the model reached high values of 0.98 and 0.97, respectively.

### 3.2. Human Readout

Table 2 depicts the classification results of the dCNN and the two readers of 60 random mammograms from the test subset. ICC between the dCNN and two readers was 0.98 (CI 95% 0.96–0.98, *p* < 0.001), representing excellent agreement. Accordingly, Kappa coefficients for the dCNN and the individual readers are summarized in Table 3. Inter-reader reliability was almost perfect in all cases (0.93–0.97). In detail, consensus between readers (0.95) and between the trained dCNN model and each reader was nearly perfect (reader 1: 0.97, reader 2: 0.93). Inter-reader reliability between classification according to the initial radiological reports and the two readers was close to perfect for reader 2 (0.95) and perfect for reader 1 (1.0). Similarly, inter-reader agreement between radiological reports and the dCNN model reached almost perfect levels (0.97). For all statistical comparisons, *p*-values were below the defined level of statistical significance (*p* < 0.001 each).

### 3.3. Colored Probability Map

The sliding window approach created high image quality predictions and was capable of analyzing the complete mammograms. An example of this technique is provided in Figure 5. The enlarged, dense axillary lymph node with round shape was correctly classified by the dCNN to class 3 (“suspicious lymph nodes”). In contrast, the adjacent lymph node appearing with normal size, oval shape, and fatty hilum as benign features was not detected. However, in this example, also, the mamilla was falsely assigned to class 3 with a lower probability. This could be explained by similar morphology of suspicious lymph nodes and mamillas, such as density and round shape.

## 4. Discussion

In this study, we successfully trained a dCNN model to detect abnormal axillary lymph nodes on mediolateral oblique mammographic images. On a cohort with high risk of malignant lymph nodes, the dCNN was able to classify breast tissue, benign lymph nodes, and suspicious lymph nodes according to the radiologic reports with similar and high diagnostic accuracy compared to experienced human readers. To the best of our knowledge, this is the first study applying a dCNN model for the detection and classification of axillary lymph nodes on mammograms.

Axillary lymph nodes represent a normal finding in a large part of MLO mammographic projections. However, if axillary lymph nodes are missing benign features, they are associated with malignancy. In case of primary breast cancer, presence or absence of metastatic disease in axillary lymph nodes is important for disease prognosis and treatment planning [15,16]. The current standard procedure for determining axillary involvement in early-stage breast cancer is performed by a sentinel lymph node biopsy (SLNB) [23,24]. Nonetheless, there is a need for non-invasive techniques complementary to SLNB for detecting lymph node metastasis. Additionally, knowledge of metastatic disease prior to surgery may alter the therapeutic management of the patient, e.g., resulting in additional whole-body staging examinations. Finally, suspicious lymph nodes might constitute the only sign of the presence of breast cancer in a mammogram, particularly in patients with dense breasts.

Therefore, it is essential to mention lymph nodes with abnormal appearance in the radiologic report, and according to our data, a dCNN can solve this task on mammograms with very high accuracy of more than 98% and excellent inter-reader agreement to human breast imaging experts. While digital mammography is very specific in the identification of suspicious lymph nodes, it is unfortunately not very sensitive as most of the axilla is pushed out of the projection and only the lower parts of the axilla are typically visualized [25,26]. Besides mammography, features suggestive of lymph node metastasis may be seen at positron emission tomography–computed tomography, sonography, and breast MRI. These modalities have shown only moderate accuracy and sensitivity in detecting axillary lymph node metastasis. Thus, complementary methods may be used to increase sensitivity and for a complete preoperative assessment of the axillary nodal status [27,28,29]. Utility of deep learning models for prediction of axillary lymph node metastasis has already been proven in ultrasound images and also in breast MRI [30,31,32,33]. In these studies, deep learning models were used for binary classification to determine if axillary lymph node metastasis was present. In comparison, our in-house-designed algorithm followed a multiclass approach to distinguish benign from malignant lymph nodes and normal tissue. Despite smaller data size, our findings confirm the feasibility of deep learning for lymph node classification with a slightly higher accuracy but using mammographic images. Although mammography has the disadvantage of potentially missing parts of the axilla, it remains by far the most frequently applied image modality in breast cancer screening programs due to its cost-effectiveness.

Artificial-intelligence-based algorithms are emerging to improve interpretation workflow in breast imaging, mostly driven by deep learning and convolutional neural networks. Since their introduction in 2012, dCNNs currently are the most powerful and most utilized machine learning algorithms for classification of radiological images [8]. When provided a very large amount of raw data, dCNNs discover features predictive of a specific outcome (in our case, “benign or suspicious lymph nodes”). Deep learning algorithms have been proven effective in detection of cancer presenting either as suspicious calcifications or masses with accuracies similar to experienced radiologists [9,34]. Schönenberger et al. reported a technique similar to our approach, but applying the dCNN successfully to classify microcalcifications on mammograms according to the ACR BI-RADS system, which is typically used by radiologists to determine clinical follow-up [11]. Parenchymal breast density is another feature and an important factor for breast cancer risk that can be depicted by deep learning models. Results show strong similarity or agreement with BI-RADS assessment made by radiologists on mammographic images [10,35] as well as on spiral breast-CT [19]. Assessment of density may even provide more accurate breast cancer risk predictions by dCNNs compared to radiologists, based on pixelwise information embedded in mammographic images that are not perceptible to the human eye [36]. Incorporation of deep learning algorithms considering several features, such as lymph nodes, breast lesions, and parenchymal density, may enable a more comprehensive report when AI is used for a second opinion.

Several limitations are included in this study. First, this was a single-center study with retrospectively collected data, including a relatively small patient population. Over a time period of nearly 10 years, only 43 mammograms from the PACS of our institution could be retrieved, which may be attributed to suboptimal reporting of suspicious lymph nodes in our radiological institutes. When searching our report (RIS) database for “suspicious lymph nodes”, the term “no suspicious lymph nodes” resulted in very many false positives, which restricted the possibilities for automatic detection of reports with the relevant findings. Due to the limited amount of data available, data augmentation had to be performed, potentially causing a bias of redundancy and a patient-based bias. Moreover, the small size of the validation dataset may be attributed to the fluctuations as seen in the accuracy and the loss function. Second, the radiological reports served as ground truth to determine whether axillary lymph nodes were labeled as pathologic or benign. Despite predefined criteria of suspicious lymph nodes and a patient cohort that had a high likelihood of malignant lymph nodes, histological proof of malignancy was missing in several cases. Third, the dCNN had only been trained to classify axillary lymph nodes on mediolateral oblique views; we did not test whether the dCNN can also detect other typical mammographic locations such as abnormal intramammary or prepectoral lymph nodes. Accordingly, it remains unclear if the dCNN achieves equal performance on craniocaudal views. Fourth, further optimization of the dCNN architecture may improve the classification accuracy, and further studies with larger datasets are warranted. However, testing different architectures is out of the scope of this proof-of-principle study. Fifth, the sliding window approach resulted in a dissimilar shape of the detected lymph node. This is explained by the different size of the sliding window (351 × 280 pixels) and the mammogram (3506 × 2800 pixels), resulting in a slight distorted shape of the lymph node once the sliding window hits the suspicious node. Furthermore, the computation time of the colormap is long in the current state. Ultimately, an output of artificial intelligence is desired that contributes to different aspects of mammographic image analysis, e.g., breast cancer detection, assessment of parenchymal density, etc. The aim is to implement deep learning models for every single aspect that reflects interpretation workflow as performed by radiologists. However, we solely focused on the aspect of detecting abnormal lymph nodes; investigating additional aspects is out of the scope of this work and should be tested in future studies.

## 5. Conclusions

In summary, we demonstrated that a dCNN can by trained to classify breast tissue, benign lymph nodes, and suspicious lymph nodes with very high accuracy similar to experienced human readers. Implementation of the dCNN may function as a standardized diagnostic tool for detection of pathological lymph nodes with high accuracy and reliability on routine mammograms, e.g., as a second opinion and decision support tool providing the expertise of a team of radiologists.

## Figures and Tables

**Figure 1 diagnostics-12-01347-f001:**
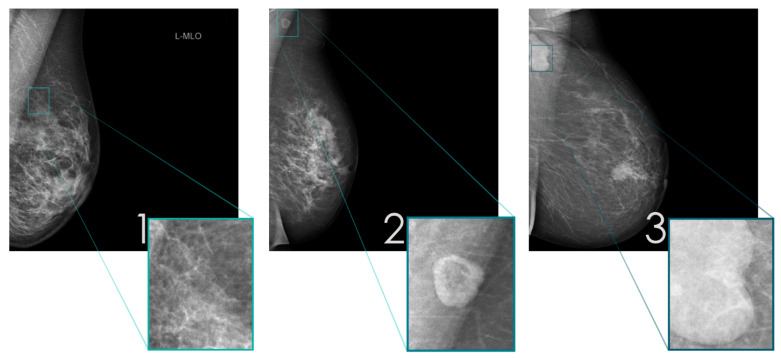
Representative examples of mammograms in MLO projection illustrating the 3 defined classes: (**1**) “breast tissue”, (**2**) “benign lymph nodes”, and (**3**) “suspicious lymph nodes”, with magnification of the ROI.

**Figure 2 diagnostics-12-01347-f002:**
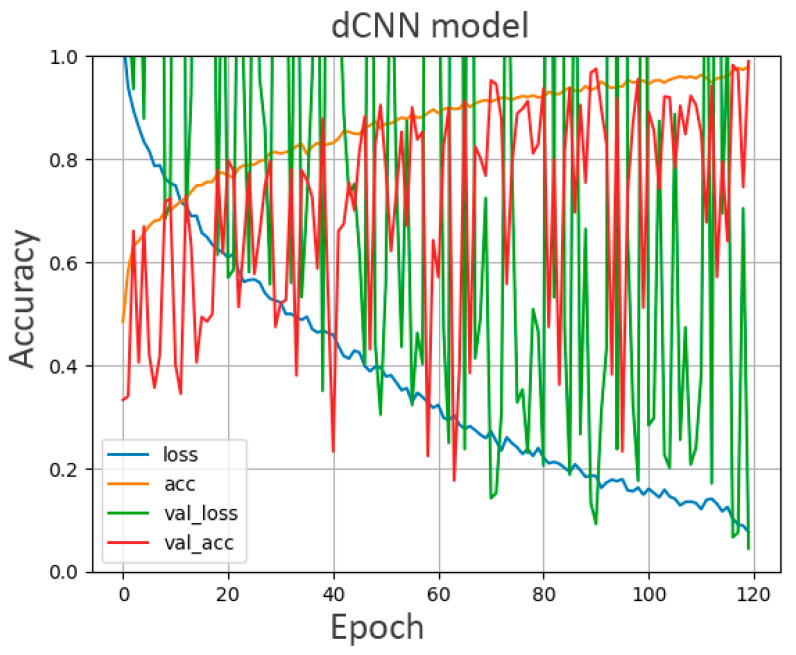
Training accuracy, validation accuracy, and loss curves for the dCNN model vs. the number of epochs for the training and validation data.

**Figure 3 diagnostics-12-01347-f003:**
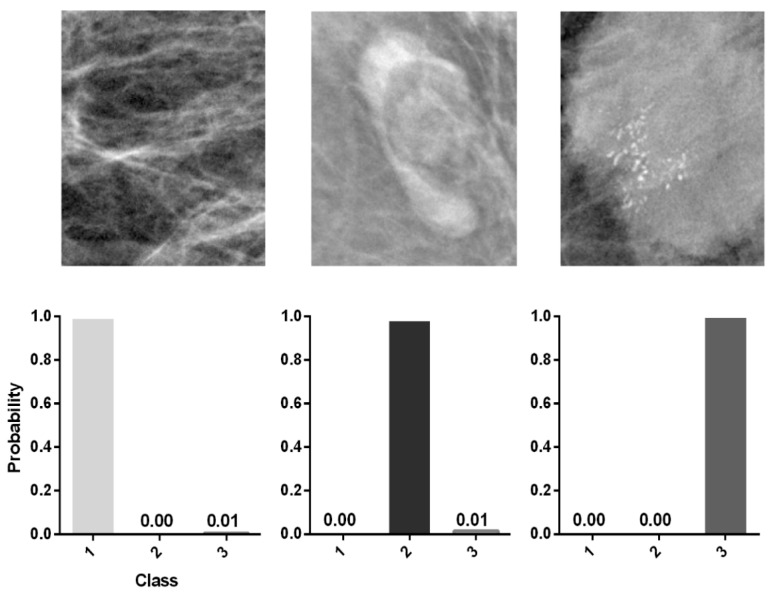
Representative mammograms of the 3 different classes (1, “breast tissue”; 2, “benign lymph nodes”; 3, “suspicious lymph nodes”) that were correctly classified by the trained dCNN according to the ground truth (radiological report).

**Figure 4 diagnostics-12-01347-f004:**
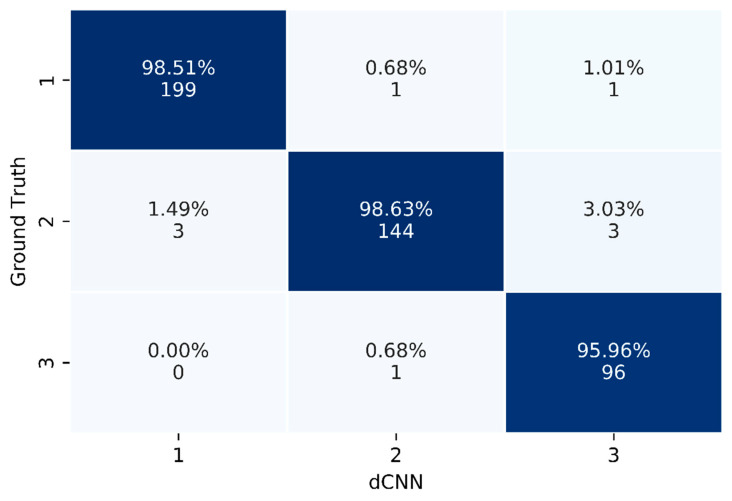
Confusion matrix of the “real world” test set calculated by the dCNN in comparison to the ground truth (radiological report). Blue marked elements highlight correctly assessed images.

**Figure 5 diagnostics-12-01347-f005:**
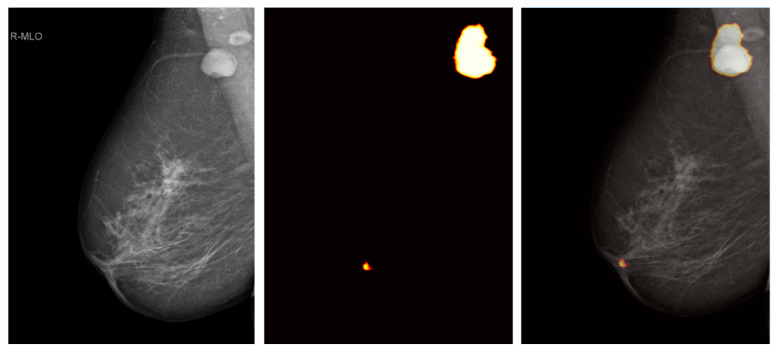
Heat colormap generated by the sliding window approach of a representative mammography in MLO projection. Left image is the original mammography, middle image the colormap, and the right image shows an overlay of both. Probability of class 3 (“suspicious lymph nodes”) is highlighted by a heat colormap.

**Table 1 diagnostics-12-01347-t001:** Number of mammograms used for training and validation of the dCNN, including number after data augmentation.

	Class		
	1	2	3
Training data	567	533	377
Augmented	2062	1926	1397

**Table 2 diagnostics-12-01347-t002:** Classification of 60 test images by the dCNN and the two readers according to the 3 classes (1, “breast tissue”; 2, “benign lymph nodes”; 3, “suspicious lymph nodes”).

	dCNN	Reader 1	Reader 2
**1**	20	20	21
**2**	19	20	20
**3**	21	20	19

**Table 3 diagnostics-12-01347-t003:** Cohen’s Kappa coefficients of the classification results between the “ground truth”, trained dCNN, and each of the two readers.

	Ground Truth	dCNN	Reader 1	Reader 2
**Ground Truth**		0.97	1	0.95
**dCNN**			0.97	0.93
**Reader 1**				0.95
**Reader 2**				

## Data Availability

The data presented in this study are available on request from the corresponding author. The data are not publicly available due to privacy restrictions.

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
