# Peer review of "Detecting Abnormal Axillary Lymph Nodes on Mammograms Using a Deep Convolutional Neural Network"

_diagnostics, 2022, doi:10.3390/diagnostics12061347_

Round 1
Reviewer 1 Report
Title: Detecting Abnormal Axillary Lymph Nodes on Mammograms Using a Deep Convolutional Neural Network
The manuscript describes a technological approach based on deep learning to analyze mammographic images devoted to the detection of abnormal axillary lymph nodes. The paper is well written, in the present form is incomplete though and therefore not worthy of publication.
General comments
- You should better discuss in the introduction why the detection of axillary lymph nodes is that complex to justify the use of advanced image processing tools. From the example, in figure 5 what are the morphological/tissuetal aspects that lead to classifying the depicted mass of the axillary lymph node as abnormal?
- What are, whether existing, traditional/alternative measurement techniques with respect to X-ray mammography for the detection of axillary lymph nodes? This should be elaborated further.
- The long-term clinical scope of the proposed system is to be better explained (screening, diagnostics, decision support system, …)
- The technical contribution of the paper is not adequately developed. The justification for the selected architecture of the network is missing. The plug-and-play use of the dCNN without any analysis or ablation study is a too simplistic approach.
- Data augmentation can be critical and the benefit is not always straightforward. According to the small dataset, the effectiveness of the convergence is questionable (Fig. 1). The conclusion about generalization property may be easily misleading.
Specific comments
- Page 2 line 78, the difference between ACR BI-RADS 6 and BI-RADS 5 is unclear
- Page 2 line 78, were all the selected patients staged with other malignancies? This is unclear.
- Page 3 line 95: why do you not use data augmentation to better balance the samples in the three classes?
- Page 4 line 139: “The approximate computation time for a standard mammography ranged between 3 and 5 hours depending on the mammogram”. what are you talking about? Training phase? Generalization phase? or what? In any case, is this time compatible with clinical practice?
- Page 5, figure 1. The trend of the validation loss and metrics is not actually what is expected for a good convergence without overfitting. This is to be verified. According to the depicted amplitudes, the convergence results are strongly dependent on the time point of stopping. Therefore, the generalization is only partially demonstrated. That is why ablation of the architecture is generally mandatory to optimize the tradeoff between over- and under-fitting.
- Page 5, Figure 5. Why does the color map produce a shape dissimilar from the node? You better should clarify the information conveyed by such an image.
Reviewer 2 Report
The abstract needs quantification. The high accuracy and kappa value indicates skewed data was analyzed. The Introduction may be enhanced. dCNN blocks may be included. F1 score and MCC may be studied. Comparison with previous studies may also be included. Conclusion has to be separated. The presence of outlier and statistical analysis may be studied.
Reviewer 3 Report
This paper presents the detection of abnormal axillary lymph nodes on mammograms using a deep convolutional neural network.
This paper is an interesting approach in the aspect of using deep convolutional neural network for detecting abnormal axillary lymph nodes in mammography images. Nevertheless, in methods based on deep learning, an important element is the learning and validation set. The effectiveness of the prediction and the thesis depends largely on the selection of criteria and the amount of data and the appropriate selection of data. The universality of the algorithm thus has some limitations.
Minor remarks:
1) I suggest describing in more detail on what basis and from what the scope of the selection of input data for the research problem posed.
2) The authors could refer to other deep learning methods and justify the choice of the presented methods.
3) The article needs minor language corrections.
Reviewer 4 Report
General Comments
While the authors have stated that this is the first paper to use a deep Convolutional Neural Network (dCNN) applied to the study of axillary lymph nodes seen on mammograms, I have some serious concerns regarding the small number of patients -- only 43 mammograms retrieved over a ten-year period. As they themselves stated (lines 271 and 272), "Due to the limited amount of data available, data augmentation had to be performed, potentially causing a bias of redundancy and a patient-based bias."
In fact, as the authors report (lines 93), the 43 mammograms were augmented using ImageDataGenerator of Keras to generate a training data set of 1477 images, with an augmented data set of 3385 images. I have grave reservations whether this is a scientifically legitimate method of increasing the input model and challenge the authors to justify that it is. I also note that in line 93 they refer to Keras version 2.0.4 while in line 111 is is Keras version 2.4.3. Which one is it?
In line 74 it is stated that "Patients' written informed consents were waived" while in lines 306 and 307 it is stated that "Written informed consent has been obtained from the patient(s) to publish the paper". These two statements are not consistent with one another.
There's also a number of other inconsistencies. For example, in the Abstract in line 11, there is reference to "1477 mammography images in MLO projection from 74 patients", whereas in lines 74 and 75, there is reference to "107 mammograms ... of 74 patients." Also in the Abstract in line 14, there is reference to "trained and validated with 5385 images" and yet in Table 2 the number of Augmented images is 3385. These sorts of inconsistencies do not give me much confidence that the authors are paying the necessary attention to detail.
In line 29, reference is made to 11.7% new cases of breast cancer and 11.4% new lung cancer cases. What are these a percentage of? The total number of all cancers, I presume, but you have to be explicit.
In lines 97 to 101, the numbers don't seem to add up. It would probably be helpful to have a diagram. I also wonder what is meant by a "real world" data set?
Specific Comments
line 34 ... settings confer substantial reduction ...
line 52 ... of artificial intelligence (AI) in medical applications.
line 54 ... input data and linking it to a ...
line 58 ... micro calcification cluster, mass and ...
line 61 ... and presents as homogeneously dense with ...
line 66 ... we tested in a retrospective cohort ...
line 86 ... has been labelled manually by one of us (FA) according to ...
line 132 ... applying a custom Python script.
line 136 ... was visualised by assigning a heat-colormap ...
line 138 ... have been generated by adjusting the ...
line 140 The computation time of 3 to 5 hours seems excessively long and perhaps warrants a comment.
line 152 Is "icon" the appropriate word to use here?
line 324 Reference 2 has no page numbers
line 365 Reference 24 has no page numbers
line 381 Reference 32 has no page numbers
Round 2
Reviewer 4 Report
I have read the response by the authors to my first review of the manuscript and I am satisfied they have responded adequately. As they now make clear in the revised manuscript, this is a proof-of-principle study and not the definitive and final word on the applicability of their model. For that to be achieved, there needs to be a significant increase in the number of patients.
Author Response
We want to thank for the constructive feedback. The proof-of-principle aspect has been additionally stressed in the final sentence of the abstract. For further research activities toward this direction, we will aim to collect bigger sample sizes.